# New Approach to Antifungal Activity of Fluconazole Incorporated into the Porous 6-Anhydro-α-l-Galacto-β-d-Galactan Structures Modified with Nanohydroxyapatite for Chronic-Wound Treatments—In Vitro Evaluation

**DOI:** 10.3390/ijms22063112

**Published:** 2021-03-18

**Authors:** Justyna Rewak-Soroczynska, Paulina Sobierajska, Sara Targonska, Agata Piecuch, Lukasz Grosman, Jaroslaw Rachuna, Slawomir Wasik, Michal Arabski, Rafal Ogorek, Rafal J. Wiglusz

**Affiliations:** 1Institute of Low Temperature and Structure Research, Polish Academy of Sciences, Okolna 2, 50-422 Wroclaw, Poland; j.rewak@intibs.pl (J.R.-S.); p.sobierajska@intibs.pl (P.S.); s.targonska@intibs.pl (S.T.); l.grosman@intibs.pl (L.G.); 2Department of Mycology and Genetics, University of Wroclaw, Przybyszewskiego 63, 51-148 Wroclaw, Poland; agata.piecuch@uwr.edu.pl (A.P.); rafal.ogorek@uwr.edu.pl (R.O.); 3Institute of Biology, Jan Kochanowski University, Uniwersytecka 7, 25-406 Kielce, Poland; jaroslaw.rachuna@gmail.com (J.R.); michal.arabski@ujk.edu.pl (M.A.); 4Institute of Physics, Jan Kochanowski University, Swietokrzyska 15, 25-406 Kielce, Poland; slawomir.wasik@ujk.edu.pl

**Keywords:** fluconazole, hydrogels, drug carriers, fungal infections, chronic wounds

## Abstract

New fluconazole-loaded, 6-Anhydro-α-l-Galacto-β-d-Galactan hydrogels incorporated with nanohydroxyapatite were prepared and their physicochemical features (XRD, X-ray Diffraction; SEM-EDS, Scanning Electron Microscopy-Energy Dispersive X-ray Spectroscopy; ATR-FTIR, Attenuated Total Reflectance-Fourier Transform Infrared Spectroscopy), fluconazole release profiles and enzymatic degradation were determined. Antifungal activity of pure fluconazole was tested using *Candida* species (*C. albicans*, *C. tropicalis*, *C. glabarata*), *Cryptococcus* species (*C. neoformans*, *C. gatti*) and *Rhodotorula* species (*R. mucilaginosa*, *R. rubra*) reference strains and clinical isolates. Standard microdilution method was applied, and fluconazole concentrations of 2–250 µg/mL were tested. Moreover, biofilm production ability of tested isolates was tested on the polystyrene surface at 28 and 37 ± 0.5 °C and measured after crystal violet staining. Strains with the highest biofilm production ability were chosen for further analysis. Confocal microscopy photographs were taken after live/dead staining of fungal suspensions incubated with tested hydrogels (with and without fluconazole). Performed analyses confirmed that polymeric hydrogels are excellent drug carriers and, when fluconazole-loaded, they may be applied as the prevention of chronic wounds fungal infection.

## 1. Introduction

Synthetic apatites in nanometric form are extensively studied in the area of biomedical applications due to their biocompatibility, high bioactivity as well as microbiological properties in vitro [1,2,3]. Since they enhance tissue regeneration, one of their application is bone grafting and teeth fillings [4] Hydroxyapatite materials in the form of hydrogels may also be used for the acceleration of skin wound healing, e.g., in a chronic or burn wound treatment [5]. Such wounds require special care and advanced therapies not only for skin regeneration but also to prevent microbial infection. Chronic wounds are an excellent gateway for the invasion of bacteria and fungi, which can adhere to the tissue and form biofilm, causing drug-resistant infection [6,7]. Loading antimicrobial compounds into hydroxyapatites is one strategy to prevent microbial growth on exposed body sites. The most frequently studied are hydroxyapatites substituted with ions and many of them have documented antimicrobial activity [8,9]. Another concept is combining hydroxyapatite and antibiotics, in which antimicrobial drug is released in the area of the application site where it inhibits microbial adhesion and growth [10].

Fungal infections, especially those caused by opportunistic pathogens, may be the repercussions of therapeutic procedures like broad-spectrum antibiotics or immunosuppressive drug therapies. Some diseases, like AIDS, also predispose subjects to fungal infections due to immune system impairment [11]. There are different types of fungal infections, regarding affected site: superficial, cutaneous, subcutaneous and systemic [11]. Fungal infections of chronic wounds have been reported in various cases. According to Dowd et al., 2010, almost one-quarter of chronic wounds may be infected by one or several fungi. The highest prevalence is attributed to yeasts-like fungi of *Candida* species, mainly *C. albicans*, but also *C. parapsilosis*, *C. glabrata* and *C. tropicalis* [6] The adhesion of fungal cells to the wounded skin may lead to the formation of the biofilm, resistant to common treatments [7].

There are several classes of antifungal agents and azoles are one of the most popular drugs. They include numerous compounds (e.g., fluconazole, itraconazole, voriconazole) with the ability to inhibit biosynthesis of ergosterol, an important component of fungal plasma membrane [12]. Azole therapy is still a first-choice treatment in many diseases caused by *C. neoformans* and common strains of *Candida* species. Due to the relatively high susceptibility of *Candida* sp. to fluconazole, as well as its safety and predictable effects in humans, it is one of preferable drugs in the treatment of diseases caused by yeasts-like fungi and yeasts, but also in prophylaxis, since its effective in 70–80% cases of *Candida* infections [13,14,15]. However, fungal resistance to common drugs should be considered before the choice of treatment. There are numerous reports describing the probability of fluconazole resistance of *C. tropicalis* and *C. albicans* suggesting that implementing therapy based on a single antifungal agent may not be sufficient; thus, there is a need to search for novel therapeutic approaches.

The combining antifungal agents (like fluconazole) and hydroxyapatite is a novel approach to prevent fungal infection, however the idea of applying dressings made of hydrogels on wound and burns was described previously. One of the widely described polymeric hydrogel is chitosan which was successfully loaded with different drugs: gentamicin sulfate, tetracycline hydrochloride, naproxen, amoxicillin, ibuprofen, nystatin and other [16]. One of the most important features of such materials is high ability of fluid absorption, what may facilitate the treatment of oozing wounds. Dressings can be made of natural (alginate, chitosan, hyaluronic acid, collagen, dextran, glucan, starch, ulvan, gelatin, pullulan) or synthetic (polyvinyl alcohol, polyacrylamide, polyethylene glycol) polymers [16,17]. Combining antifungal agent with drug carrier (hydrogel) provides stable treatment in the site of its administrations and prolongs drug release from the dressing. 

Moreover, generally such materials are biocompatible, noncytotoxic and nonimmunogenic [16]. Since the wet environment inside the wound promotes microbial growth, the application of such wound dressing would not only enhance wound healing and skin regeneration, but also block microbial adhesion to the wound and prevent host tissue invasion [18]. Thus, the main goal of this research is to estimate antifungal activity of prepared hydroxyapatite hydrogels loaded with fluconazole against *Candida* sp. (*C. albicans*, *C. tropicalis*, *C. glabrata*), *Cryptococcus* sp. (*C. neoformans*, *C. gatti*) and *Rhodotorula* sp. (*R. mucilaginosa*, *R. rubra*) by evaluation of fungal adhesion to hydrogels. The newly synthesized materials may be applied in the future as chronic wound dressings.

## 2. Results

### 2.1. Physicochemical Characterisation of the Obtained Hydrogels

The crystal structure of investigated hydrogels was determined by X-ray powder diffraction (XRPD) and presented in Figure 1. The hexagonal (P6_3_/m space group) structure of nanocrystalline hydroxyapatite was confirmed. The diffraction peaks at 2θ angles equal to 25.94; 31.85; 32.19; 32.93 and 34.10 were indexed to hydroxyapatite host lattice (ICSD-26204) [19]. The peaks corresponding to the hydroxyapatite structure were also identified in the patterns of nHAp/6-Anhydro-α-l-Galacto-β-d-Galactan hydrogel as well as Fluconazole/nHAp/6-Anhydro-α-l-Galacto-β-d-Galactan hydrogel. The XRPD patterns of Fluconazole/nHAp/6-Anhydro-α-l-Galacto-β-d-Galactan hydrogel and Fluconazole/6-Anhydro-α-l-Galacto-β-d-Galactan hydrogel have displayed characteristic diffraction peaks of fluconazole. The most intense peaks are located at the 2θ at 9.03; 10.05; 16.19; 16.66; 20.10; 25.56. Two sharp peaks at 21.26 and 23.65 derived from pure 6-Anhydro-α-l-Galacto-β-d-Galactan hydrogel.

For the chemical characterization of obtained materials, the Fourier-Transform Infrared spectra were recorded (see Figure 2). The typical line ascribed to 6-Anhydro-α-l-Galacto-β-d-Galactan hydrogel are detected at 930, 1028 and 1073 cm^−1^ attributed to the C-O vibrational modes [20]. Two absorption bands that appeared at 2850 and 2917 cm^−1^ are attributed to the symmetric and antisymmetric stretching vibration of -CH_2_ groups. The broad band located in the range of 3700 to 3020 cm^−1^ is related to water molecules. The spectrum of fluconazole is built by a number of lines, showing features typical to organic compounds. The existence of 1,2,4-trisubstituted benzene ring is proved by the appearance of lines at 1620, 674 and 524 cm^−1^ associated with aromatic stretching mode νC=C and benzyl ring deformation [21]. The lines at 844 and 960 cm^−1^ are ascribed to the out-of-plane ring bending vibration of -CH group. The in-plane ring -CH bending vibration and the C-F stretch are overlapped by each other and are detected at 1270 cm^−1^ [22]. Furthermore, another line at 1109 cm^−1^ could be assigned to the C-F stretching mode. The most characteristic lines that belonged to hydroxyapatite molecules vibration are covered by C-O vibration mode. The *ν_4_*(PO_4_^3−^) triply degenerated vibrations are detected at 602 and 563 cm^−1^. The lines attributed to the stretching vibration of OH^−^ groups are located at 633 cm^−1^ [23].

The chemical compositions of obtained hydrogels were confirmed by SEM-EDS analysis (Figure 3a–c). Representative spectrum of fluconazole/nHAp/6-Anhydro-α-l-Galacto-β-d-Galactan hydrogel is presented in Figure 3c. Based on the wt% values gathered in the table (inset), the Ca/P molar ratio was calculated to be 1.68, which is similar to the theoretical value (1.67) for hydroxyapatite. Other elements were detected as well. Carbon (C) derived from the matrix and drug and oxygen (O) is a part of all components but F element is only contained in the drug fluconazole. Based on the elemental mapping (Ca and P maps), the homogeneous distribution of the hydroxyapatite nanocrystals in the hydrogel matrix was found (see Figure 3b). Moreover, it can be seen that the drug (F and merged maps) was also quite evenly dispersed. However, it should be noted that both compounds, especially hydroxyapatite, show a tendency to agglomerate, as shown in Figure 4.

The morphology of pure 6-Anhydro-α-l-Galacto-β-d-Galactan (a) and filled with Fluconazole (b) or hydroxyapatite (c) as well as 6-Anhydro-α-l-Galacto-β-d-Galactan with dispersed fluconazole and nHAp (d) have been presented in Figure 4. It is clearly seen that the surface of hydrogel matrix with and without the drug has a smooth flaps-like morphology arranged in layers, whereas the addition of nHAp resulted in the creation of rough morphology. It was also found that nHAp clumped into smaller regions of the hydrogel matrix (c,d). A similar observation has been made in our previous research [24]. We noticed that nHAp-modified hydrogel increases the proliferative activity and viability of human multipotent stromal cells (hASCs), as well as reduces markers of oxidative stress. Such a surface modification should improve the bioavailability of the material for biological tissues. Moreover, it may affect drug release and interactions with various microorganisms such as fungal planktonic cell what was further investigated in this work.

### 2.2. Fluconazole Release from Fluconazole/nHAp/6-Anhydro-α-l-Galacto-β-d-Galactan Hydrogel

Drug release profiles of the two examined hydrogels are shown in Figure 5. As expected, almost complete release was observed in phosphate buffer (pH = 6.8), which is related to the BCS (Biopharmaceutics Classification System) classification of fluconazole. Fluconazole belongs to BCS class 1—High solubility compound in the entire physiological pH range. Although EMA Guidelines and European Pharmacopoeia recommended 100 rpm rotation speed for basket method, 50 rpm agitation speed was selected for this study, because it seems to be more discriminatory and more suitable for the comparison of the hydrogels. Low RSD values between samples confirm that the study conditions were selected correctly. In Figure 5 it is clearly seen that after 1 h of drug release the plateau of the drug release profile has been reached for two examined hydrogels and there were no statistically significant differences between the hydrogel with and without nHAp.

Moreover, the kinetics of fluconazole release from 6-Anhydro-α-l-Galacto-β-d-Galactan hydrogel modified with nanohydroxyapatite (Fluconazole/nHAp/6-Anhydro-α-l-Galacto-β-d-Galactan) was determined by the laser interferometry. This technique gives the possibility of the drug transport measurement in the physiological-like conditions. In detail, laser interferometry technique was used to measure fluconazole release from the hydrogel in real-time in an unmixed system, in comparison to the HPLC study (Section 2.2). It seems that this interferometric study complements the biophysical characteristic of 6-Anhydro-α-l-Galacto-β-d-Galactan hydrogel modified with nanohydroxyapatite and fluconazole as biomaterial dedicated to clinical application.

Figure 6 shows that 0.93 mg of fluconazole was released after 54 min from Fluconazole/nHAp/6-Anhydro-α-l-Galacto-β-d-Galactan hydrogel and for the next 6 min the level of the released drug generally did not change (S.D. = 0.02 mg). It means that the concentration of fluconazole was levelled at hydrogel–water interface according to diffusion phenomena. The used block of Fluconazole/nHAp/6-Anhydro-α-l-Galacto-β-d-Galactan hydrogel (7.8 mg) contained c.a. 2 mg of fluconazole. Thus, after 1 h the effective transport of the drug was detected, according to laws of diffusion in an unmixed system at 37 °C. It confirms that the transport properties of fluconazole from the nHAp/6-Anhydro-α-l-Galacto-β-d-Galactan hydrogel are great, as was observed in the HPLC study.

### 2.3. nHAp/6-Anhydro-α-l-Galacto-β-d-Galactan Hydrogel Degradation by Lysozyme

The degradation of nHAp/6-Anhydro-α-l-Galacto-β-d-Galactan hydrogel by lysozyme was analysed by laser interferometry. Figure 7 shows the amount of hydrogel degradation products (HDP) in arbitrary units (a.u.). Figure 7A shows that c.a. 0.027 a.u. of HDP was released to the water phase for each 2 min, after the first 4 min. measured in a short time period. The kinetics of this releasing process in the function of time are presented in Figure 7B. In summary, c.a. 2 a.u. of HDP was transported to the water phase after 80 min; this increase seemed to be linear.

Additionally, the spatiotemporal distributions of refraction index changes for HDP after incubation with lysozyme are presented in Figure 8. These distributions reflect the spatiotemporal distribution of HDP concentration, presented in arbitrary units. These profiles show that a greats amount of HDP is near to the nHAp/6-Anhydro-α-l-Galacto-β-d-Galactan hydrogel–water interface. For example, after 80 min the concentration of degradation products at point x = 0 equals 2.8 a.u. Moreover, the profiles allowed to determine the distance from the hydrogel to each transported product. For example, HDP are transported at a distance of 4.209 mm from the nHAp/6-Anhydro-α-l-Galacto-β-d-Galactan hydrogel surface (x = 0).

### 2.4. Antifungal Activity

#### 2.4.1. Minimal Inhibitory Concentrations

To investigate antifungal activity of fluconazole minimal inhibitory concentrations were determined against various yeast strains (clinical isolates and reference strains). MIC values are listed in the Table 1. The most sensitive to fluconazole were both tested *C. albicans* strains and three clinical isolates of *C. glabrata* (133, 183 and 327), for which MICs were lower than 2 µg/mL. High susceptibility was also exhibited by the reference strain of *C. tropicalis* (MIC of 4 µg/mL). The highest MIC values were observed in the case of *R. rubra*, *R. mucilaginosa* (above 250 µg/mL) and the reference strain of *C. glabrata* (250 µg/mL) Among tested *Cryptococcus* strains, moderate susceptibility to fluconazole was noted, ranging from 15.6 µg/mL (*C. neoformans* No. 8) to 125 µg/mL (*C. gatti* CBS 12754).

#### 2.4.2. Biofilm Formation by Tested Strains

For the selection of suitable conditions for biofilm formation, the ability to produce biofilm by yeast strains at 28 ± 0.5 °C (optimal for yeast growth) and 37 ± 0.5 °C (temperature of human body) was compared and presented in Figure 9. No significant differences in biofilm formation at 28 ± 0.5 and 37 ± 0.5 °C were observed for all tested strains. Absorbances ranged from 0.15 (*C. grabrata* 183) to 2.87 (*C. glabrata* 133). The highest biofilm production was exhibited by *C. glabrata* 133 and 327 (vaginal isolates).

#### 2.4.3. Biofilm Formation on the Surface of Tested Materials and Influence on Fungal Planktonic Cells

For the investigation whether the yeasts are able to form biofilm on HAp materials and survive the presence of released fluconazole, microscopic observations in confocal microscope with LIVE/DEAD staining were performed. Additionally, the absorbance of the medium used for cultivation of biofilms was measured to evaluate the survival of non-adhered cells (Figure 10). For this purpose, four fungal strains with the highest ability to produce biofilm and the clinical significance were selected.

The microscopic images (Figure 11) suggest that the most preferable surface for biofilm formation is 6-Anhydro-α-l-Galacto-β-d-Galactan combined with hydroxyapatite and pure 6-Anhydro-α-l-Galacto-β-d-Galactan is less prone to be colonised by tested fungi. The cells within the biofilms remained alive despite the significant reduction in the absorbance of the medium—Fluconazole released from HAp reduced fungal growth almost completely (Figure 10). Only *R. mucilaginosa* biofilm was susceptible for fluconazole incorporated into hydrogels to some extent as red-fluorescent (dead) cells were visible in every image (Figure 11). Some changes in biofilm morphology are visible in the case of 6-Anhydro-α-l-Galacto-β-d-Galactan combined with fluconazole. Aggregates of adherent fungal cells were formed on the material. This effect was, however, no longer visible when the combination of 6-Anhydro-α-l-Galacto-β-d-Galactan, HAp and fluconazole was applied.

## 3. Discussion

Synthetic hydroxyapatites with various modified structures are widely studied towards their application in bone replacement and as a dental material, but also as a potential dressing in chronic wound treatment [25,26,27]. Antimicrobial properties of hydroxyapatite material are desirable since the implantation carries a risk of an infection. In the case of chronic or burn wound treatment, the wounded tissue is vulnerable to microbial invasion [28]. The antibacterial activity of various hydroxyapatites with modified structures, e.g., with metal ion-doping, is well documented [29].

Although the antifungal properties of some hydroxyapatites are also described, the data on this issue is rather scarce [30]. The focus on an antibacterial activity seems to be obvious, since bacteria are the main reason of nosocomial and wound infections [28]. Bacterial cells, once adhered to the wounded tissue, may develop into the biofilm, what may severely interrupt wound treatment. It was proven that wound infections caused by biofilms delay chronic wound healing e.g., by dispersing from the biofilm and to other sites and elevation of wound moisture. Moreover, the treatment of the chronic wound infections is often impeded due to the higher tolerance of biofilms to antibiotics, thus, other therapies, such as the application of nanoparticles, must be introduced [31]. On the other hand, the role of yeasts in the disease development cannot be omitted as they are not only common inhabitant of human skin and mucosa, but also might be found in hospital environments, where they can adhere to the surfaces of e.g., surgical devices [32]. Once adhered to the surface, yeasts may form a biofilm, a three-dimensional structure, composed of cells and extracellular matrix, and grow in such community. Fungi are also a part of chronic wound microbiome where they interfere with the healing process e.g., by causing wound necrosis. The biofilm formation by fungi in chronic wounds, such as diabetic-foot ulcers, was also observed, what can impede antibiotic treatment [33]. Azoles, among available antifungals, are one of the most frequently used drugs in the treatment of fungal infections [34]. Thus, hydroxyapatite-based hydrogels combined with fluconazole were synthesized as a potential alternative for the dressing in the treatment of wounds vulnerable for the yeast infection.

In this work, we showed that there are no significant differences in the release of the fluconazole from the 6-Anhydro-α-l-Galacto-β-d-Galactan matrix and in the presence of hydroxyapatite. The results showed a nHAp-fluconazole interaction without the participation of strong chemical bonds. Nevertheless, hydroxyapatite did not block the release of the drug into the surrounding environment, which is important for the biological activity. The release of fluconazole from the hydroxyapatite material should prevent fungi from wound invasion, however the susceptibility of yeasts and yeast-like fungi to this drug vary not only between the species, but also between the strains [35]. As it was shown in present studies the most susceptible to fluconazole were *C. albicans* and some of the *C. glabrata* strains, both associated with the wound fungal microbiome [36]. *C. albicans* MIC values were very low, but previous research conducted on the wide spectrum of isolates revealed broad diversity in fluconazole susceptibility; however, the vast majority of tested strains were highly susceptible [35,37,38].

Other tested strains exhibited high or moderate resistance, which corresponds to the results obtained previously by Pelletier et al., 2002 and Yang et al., 2004. Among 27 isolates of *C. tropicalis*, 23 were highly or moderately susceptible [35,38].

Interestingly, *Rhodotorula* sp., which is associated with wound infections, but also an inhabitant of the hospital environment, exhibited the highest resistance to fluconazole [36,39]. These results are in agreement with previous research suggesting that this genus is not the target for azole drugs [39,40,41,42].

The ability to produce biofilm by microorganisms is one of the virulence factors, facilitating survival in human body. Fungi may adhere to medical devices (such as catheters or implants) but also to the host tissue. The formation of biofilm leads to the increased resistance to treatment, due to (among others) extracellular polymeric matrix, impeding drug penetration. The ability to produce biofilm by tested fungi varied between the species, with the highest value observed for two strains of *C. glabrata* and was independent on the temperature of incubation. Moreover, the biofilm formation was also strain-dependent, since the greatest biofilm was produced by vaginal isolates of *C. glabrata* and the weakest by cervical canal isolates, suggesting a possible role of biofilm in pathogenicity and invasion of those sites.

*C. albicans* and non-albicans species vary in the ability to form biofilm. Thein et al., observed in 2007 that dynamic incubation promotes biofilm formation [43]. As it was described by Kucharíková et al., 2011 the ability to produce biofilm by *C. albicans* and *C. glabrata* is influenced not only by the temperature of incubation but also by pH and the composition of the medium [44]. Our experiments were conducted in SD minimal medium to preserve equal conditions for both MIC and biofilm production testing. However, results obtained previously indicate that RPMI 1640 medium may be more suitable for biofilm growth because its components promote multilayer structure formation [44].

*Cryptococcus* isolates formed biofilm at rather low level, especially in the case of *C. gatti*; however, for *C. neoformans*, particularly at 28 °C, the obtained results were moderate. Martinez and Casadevall, 2007 revealed that intensity of biofilm production by *C. neoformans* strongly depends on several factors like temperature of incubation, surface conditioning or pH of the medium [45].

As it was shown both *Rhodotorula* isolates can be classified as medium biofilm producers what follows the results obtained by Nunes et al., 2013 who proved that biofilm produced by clinical isolates is greater than by the reference strains [39].

The yeast strains with the higher ability to form biofilm were investigated towards biofilm formation on hydroxyapatite materials. The increased resistance of fungal biofilms to fluconazole in comparison to planktonic cells is well documented [46]. Similarly, present results showed significant differences in the survival of yeasts incubated in the presence of HAp material. As it was observed, the absorbance of the medium measured after the incubation of fungi with the fragments of hydroxyapatite hydrogels combined with fluconazole was much lower comparing to the control and uncombined HAp. Antifungal effects of nanoparticles (NPs) containing antimicrobial agents are a common approach for infection treatment. Silver NPs are one of the most frequently tested against microorganisms with well documented antifungal effects [47,48]. Additionally, combining silver NPs with ketoconazole, another azole drug, was also investigated and showed high inhibition rate when applied as gel [49]. Synergistic inhibition effect was also proved for silver NPs combined with epoxiconazole [50].

The microscopic observations of biofilms formed on those materials showed no or minor lethal effect on yeast cells. These results might suggest that fluconazole released from the hydrogels killed most of the planktonic cells, but not the adhered ones. The adhesion of yeasts to hydroxyapatites is often facilitated by various factors. *C. albicans* are known for the adhesion to dental material composed, among others, of HAp via electrostatic interaction [51]. Additionally, the porous surface of hydrogels might facilitate cell–surface interactions. The structure of biofilm formed on the hydrogels also varied dependently on the material composition. The combination of fluconazole and 6-Anhydro-α-l-Galacto-β-d-Galactan changed the morphology of the biofilm, causing yeast cells to aggregate into the clusters. Such an effect was observed previously in the studies on fluconazole effect on *C. glabrata* biofilms. The authors suggested enhanced secretion of polysaccharides and proteins, lowering the hydration of the biofilm [46]. This effect was not observed, however, when the combination of fluconazole, 6-Anhydro-α-l-Galacto-β-d-Galactan and HAp was applied, suggesting an inhibiting role of HAp in cell clustering. 

Similar conclusions were made in the studies on bacterial biofilms formed on hydroxyapatites [52]. Interestingly, some lethal effect of hydrogels with fluconazole on *R. mucilaginosa* biofilm was observed, in the contradiction to the results obtained in the experiment with the activity of pure fluconazole on planktonic cells. This was especially notable in the case of hydrogels based on 6-Anhydro-α-l-Galacto-β-d-Galactan, HAp and fluconazole. Further studies on this species are required to understand which factors are responsible for such observations. Obtained results suggest that hydroxyapatite-based hydrogels may be used as fluconazole carrier and they can be applied as antifungal treatment in sites with high fungal infection probability, like chronic or burn wounds. Thattaruparambil Raveendran et al., 2019 previously evaluated biological activity of chitosan bandages loaded with fluconazole and fibrin-nanoparticles and proved its antifungal activity on *C. albicans* strains. Moreover, a cytotoxicity test performed on fibroblast cell line confirmed its safety [53]. Similar results were obtained for polyvinyl alcohol-based fluconazole-loaded material [54].

## 4. Materials and Methods

### 4.1. Preparation of Nanocrystalline Hydroxyapatite

The nanocrystalline powder of hydroxyapatite was prepared by the precipitation method. Analytical grade Ca(NO_3_)_2_∙4H_2_O (99% Acros Organics, Schwerte, Germany), NH_4_H_2_PO_4_ (99,995% Alfa Aesar, Haverhill, MA, USA) and NH_4_OH (99% Avantor Poland, Gliwice, Poland) for pH adjustment were used. At first, the stoichiometric amount of calcium nitrate was dissolved in deionized water. Then, the suitable amount of ammonium dihydrogen phosphate was added to the previous mixture leading to the fast precipitation of the of the intermediate product. The pH of the dispersion was modulated to 9–10 by addition of ammonia. The reaction mixture was heated and stirred for 1 h. Subsequently, the obtained product was washed several times with de-ionized water and dried at 70 °C for 24 h. Finally, the product was heat treated at 450 °C for 3 h.

### 4.2. Preparation of 3,6-Anhydro-α-l-Galacto-β-d-Galactan Samples Incorporated with HydroxYapatite and Fluconazole

Pure 6-Anhydro-α-l-Galacto-β-d-Galactan and 6-Anhydro-α-l-Galacto-β-d-Galactan hydrogels: (i) filled with hydroxyapatite, (ii) loaded with fluconazole and (iii) hydrogel incorporated with hydroxyapatite and fluconazole were prepared. In order to prepare Fluconazole/nHAp/6-Anhydro-α-l-Galacto-β-d-Galactan hydrogel the following procedure was applied. At first, 3,6-Anhydro-α-l-Galacto-β-d-Galactan (BioMaxima Spółka Akcyjna, Lublin, Poland) was dissolved in deionized water by heating at around 70 °C and stirring for 1 h. Subsequently, nanohydroxyapatite (nHAp) and fluconazole (2-(2,4-difluorophenyl)-1,3-bis(1,2,4-triazol-1-yl)propan-2-ol) were added. The mixture was stirred until complete homogenization. Afterward, the glycerine was added. The obtained suspension was cooled to reach room temperature and transferred to the petri dish. The obtained hydrogel was transferred to the lyophilization process to obtain a porous gel form. The same methodology was used for other hydrogels.

### 4.3. Materials Characterization

The X-ray powder diffractograms (XRD) were recorded on a PANalytical X’Pert Pro X-ray diffractometer (Malvern Panalytical Ltd., Malvern, UK) equipped with Ni-filtered Cu Kα1 radiation (Kα1 = 1.54060 Å, V = 40 kV, I = 30 mA). FT-IR spectra were recorded using a Thermo Scientific Nicolet iS50 FT-IR spectrometer (Waltham, MA, USA) over the wavenumbers 4000–500 cm^−1^ (spectral resolution was set to 4 cm^−1^) The ATR (Attenuated Total Reflection) spectra were recorded using Nicolet iS50 FT-IR (Thermo Scientific) spectrometer equipped with an Automated Beamsplitter exchange system (iS50 ABX containing DLaTGS KBr detector), built-in all reflective diamond ATR module (iS50 ATR), Thermo Scientific PolarisTM and HeNe laser as an IR radiation source. Spectral resolution was set to 4 cm^−1^. The microstructure of obtained hydrogels and elemental analysis together with the mapping of elements were carried out using a scanning electron microscope FEI Nova NanoSEM 230 (FEI Company, Hillsboro, OR, USA) equipped with an energy dispersive spectrometer (EDAX PegasusXM4,FEI Nova NanoSEM 230) and operating at an acceleration voltage in the range of 3.0–15.0 kV and spot 2.5–3.0 were observed.

### 4.4. Fluconazole Assay

The assay of fluconazole was determined by the HPLC method (Agilent 1260 Infinity system, Agilent Technologies Inc., Santa Clara, CA, USA) using reverse phase column C18 (3.5 µm particle size, 75 mm, 4.6 mm, Zorbax, Agilent Technologies Inc., Santa Clara, CA, USA) at 30 °C. The mobile phase was a mixture of acetonitrile and water adjusted to pH = 2.5 20:80 (*v*/*v*/*v*). The autosampler temperature was set at room temperature. The injection volume was 50 µL, flow rate 2 mL/min, and run time 2 min. Sample solution was prepared in 0.1 M HCl and filtered through 0.2 µm syringe filter into HPLC vial. The content of fluconazole was determined at 260 nm and amounted to be 9.5 mg of fluconazole in 25 mg of Fluconazole/6-Anhydro-α-l-Galacto-β-d-Galactan hydrogel and 6.6 mg of fluconazole in 25 mg of Fluconazole/nHAp/6-Anhydro-α-l-Galacto-β-d-Galactan.

### 4.5. Drug Release Testing

Drug release profiles were evaluated according to the European Pharmacopoeia apparatus 1, basket method (SR8 PLUS bath with autosampler, Teledyne Hanson Research, Los Angeles, CA, USA). In all experiments 500 mL of phosphate buffer pH = 6.8 at 37.0 ± 0.5 °C as a medium was used. Basket rotation speed was 50 rpm. Aliquots of 5 mL were withdrawn at predetermined time intervals (0,5; 1; 2; 3; 4; 5; 6h) and replaced with an equal volume of fresh medium to maintain a constant total volume in each vessel. Samples were collected in HPLC vials. The concentration of each sample was determined using the HPLC method.

### 4.6. Laser Interferometry Method Measurements

The diffusion properties of fluconazole, releasing from Fluconazole/nHAp/6-Anhydro-α-l-Galacto-β-d-Galactan hydrogel to ultra-pure water at 37 °C was determined experimentally using the laser interferometry method [55,56]. The system consists of a two-beam Mach–Zehnder interferometer with an He-Ne laser type HN 40P (Zeiss, Oberkochen, Germany), a cuvette (internal dimensions: 70 mm high, 10 mm wide, optical path length: 7 mm) made with optical glass of high uniformity, a TV-CCD camera, and a computer with software for the acquisition and processing of interference images (interferograms). The lyophilized hydroxyapatite block after incubation with fluconazole was placed at the bottom of the cuvette which was then filled with the ultra-pure water (initial substance concentration C_0_ = 0). The transport properties of fluconazole from Fluconazole/nHAp/6-Anhydro-α-l-Galacto-β-d-Galactan hydrogel to water were measured using above laser interferometry system on the basis of obtained interferograms. A computer image-processing system, complete with a dedicated software, enables mathematical analysis interferograms shown on the system screen. Moreover, taking a series of pictures of interference images in time and conducting a mathematical analysis thereof enables a quantitative analysis of real-time release kinetics and concentration distribution of fluconazole in near-hydroxyapatite surface fields. 

The interferograms, which appear due to the interference of laser beams, were determined by the refraction coefficient of the solute which in turn depends on substance concentration. When the solute is uniform, the interference fringes are straight, and they bend when a concentration gradient appears. The basis for determination of the spatiotemporal substance concentration distribution (e.g., concentration profile) C(*x*,*t*) is the proportionality coefficient between changes in substance concentration ΔC(*x*,*t*) and corresponding changes of the solution’s refractive index ∆*n*(*x*,*t*). The correlation between concentration and refractive index of water solutions was refractometrically determined. The concentration profile *C*(*x*,*t*) was determined by the deviation *d*(*x*,*t*) of the fringes from their straight course. Since the concentration *C*(*x*,*t*) and the refraction coefficient are assumed to be linear [57], the equation is:(1)$$C\left(x,t\right)=C_{0}+a\Delta n=C_{0}+a\frac{\lambda d\left(x,t\right)}{hf}$$
where *C*(*x*,*t*) denotes the concentration of fuconazole at a point situated at the distance *x* from the Fluconazole/nHAp/6-Anhydro-α-l-Galacto-β-d-Galactan hydrogel–water interface; *C*_0_ is the initial substance concentration (C_0_ = 0); *a* is the proportionality constant between the concentration and the refraction index (*a* = 4373.3 mg/mL for the fluconazole aqueous solution); λ is the wavelength of the laser light (632.8 nm); *h* is the distance between the fringes in the field where they are straight lines; and *f* is the thickness of the solution layer in the measurement cuvette. By recording the interferograms over a given time interval one can reconstruct the concentration profiles at different times. The interferograms were recorded from 120 to 4800 s with a time interval of Δ*t* = 120 s and the concentration profiles for each interferogram were reconstructed.

On the basis of concentration profiles, one can determined the transport parameters substance quantity, such as substance quantity after time *t* (*N*(*t*)):(2)$$N\left(t\right)=S\int_{0}^{\delta }C\left(x,t\right)dx$$
where: *S* is the surface of Fluconazole/nHAp/6-Anhydro-α-l-Galacto-β-d-Galactan hydrogel–water interface (*S* = 7 × 10^−5^ m^2^), and *δ* is the concentration boundary layer thickness defined as the distance between the Fluconazole/nHAp/6-Anhydro-α-l-Galacto-β-d-Galactan hydrogel–water interface and the point at which the deviation of the interference fringe from its straight line run amounts to 10% of the fringe thickness.

The laser interferometry method was also used in the studies of nHAp/6-Anhydro-α-l-Galacto-β-d-Galactan hydrogel degradation by 1 mg/mL lysozyme at 37 °C for 18 h. For this purpose, the total amount of hydrogel degradation products (HDP) released from the nHAp/6-Anhydro-α-l-Galacto-β-d-Galactan hydrogel nHAp treated with lysozyme was analysed. Since the proportionality constant *a* between the concentration and the refractive index for the HDP released in this experiment is unknown, the sums of the changes of refraction index (*SCRI*) of the solute were calculated:(3)$$SCRI\left(t\right)=S\int_{0}^{\delta }\Delta n\left(x,t\right)dx$$

The *SCRI*(*t*) reflects the time dependence of the total amount of HDP (in arbitrary units) released from the nHAp/6-Anhydro-α-l-Galacto-β-d-Galactan hydrogel to the water phase.

### 4.7. Antifungal Activity

#### 4.7.1. Minimal Inhibitory Concentrations of Fluconazole

Antifungal activity of pure fluconazole was tested using broad spectrum of fungal strains, often reported as opportunistic or pathogenic: reference strains (*C. albicans* ATCC 90028, *C. tropicalis* ATCC 750, *C. glabrata* ATCC 90030, *C. neoformans* H99 *C. gatti* R265 and *R. mucilaginosa* JHM 18459) and clinical isolates (*C. albicans* R41/R42 (blood), *C. tropicalis* 11MD/2017 (oral cavity), *C. glabrata* R253/R254 (blood), *C. glabrata* 133 (vagina), *C. glabrata* 137 (vagina), *C glabrata* 183 (cervical canal), *C. glabrata* 260(cervical canal), *C. glabrata* 327 (vagina), *R. rubra* 12MD/2017(toenails), *C. neoformans* no. 8 and *C. gatti* CBS 12754. Tested strains came from the collection of the Department of Mycology and Genetics, University of Wroclaw and Department of Microbiology, Faculty of Medicine Wroclaw Medical University.

Standard microdilution method in SD (Synthetic Defined; 6.7 g/l Yeast Nitrogen Base w/o aminoacids (Difco; MERCK, St. Louis, Missouri, USA) and 20 g/l glucose anhydrous pure p.a. (CHEMPUR, Piekary Śląskie, Polska) medium, was used to determine MIC values of fluconazole. Tested fungi were incubated in YPD broth (Yeast Extract Peptone Dextrose; A&A Biotechnology, Gdynia, Polska) with shaking (100 rpm) at 28 ± 0.5 °C overnight for *Candida* strains or 48 h for *Rhodotorula* and *Cryptococcus*). Then, cultures were centrifuged and obtained fungal pellets were suspended in SD medium to obtain optical density of 0.5 in McFarland scale. The suspensions were diluted 250× in SD medium.

The final concentration of fluconazole ranged between 2 to 250 µg/mL and it was obtained by a series of two-fold dilutions of 500 µg/mL fluconazole in SD medium mixed with fungal suspension in 1:1 ratio. All samples were transferred in triplicate to a 96-well plate and incubated at 28 ± 0.5 °C with shaking for 24 h (*Candida*) or 48 h (*Rhodotorula*, *Cryptococcus*). Then, optical density was measured (λ = 600 nm) using microplate reader (Varioskan LUX, Thermo Fisher Scientific; Waltham, MA, USA) and MICs (minimal inhibitory concentrations) corresponded to fluconazole concentration that completely inhibited fungal growth (no visible density).

#### 4.7.2. Assessment of Biofilm Production Ability

To evaluate an ability of the biofilm formation by tested strains, fungi were incubated as described above and suspensions (OD_600_ = 0.8) were prepared in SD medium and placed in a 96-well plate in 8 repetitions (200 µL in each well). Two identical plates were prepared: one of them was incubated at 28 ± 0.5 °C and the other at 37 ± 0.5 °C, both with shaking (100 rpm) for 24 h. Then, the wells were rinsed with distilled water three times and left for 15 min to dry at the room temperature. Then, 200 µL of 0.1% crystal violet (CV) (“AQUA-MED” ZPAM—KOLASA sp. j.; Łódź, Poland) was added to each well and left for 15 min, the wells were then rinsed with distilled water three times. Finally, 200 µL of 95% ethanol was added to each well to dissolve bound CV and the absorbance was measured using microplate reader (Varioskan LUX, Thermo Fisher Scientific) (λ = 595 nm).

#### 4.7.3. Confocal Microscopy

For this experiment, fungal strains with the highest biofilm production were selected: *C. albicans* ATCC 90028, *C. tropicalis* ATCC 750, *C. glabrata* 133, and *R. mucilaginosa* JHM 18459.

Four types of hydrogels (pure, with hydroxyapatite, with fluconazole as well as with hydroxyapatite and fluconazole) were used to compare biofilm formation ability. Material was aseptically cut into pieces and put into 24-well plate. The masses of particular samples were as follows: 10 mg of pure 6-Anhydro-α-l-Galacto-β-d-Galactan, 20 mg of 6-Anhydro-α-l-Galacto-β-d-Galactan with apatite, 20 mg of 6-Anhydro-α-l-Galacto-β-d-Galactan with fluconazole and 30 mg of 6-Anhydro-α-l-Galacto-β-d-Galactan with hydroxyapatite and fluconazole in order to maintain equal concentration of the particular components in each. Fungal suspensions were prepared as described above and 1 mL of each strain suspension was added to the wells. The plate was incubated for 24 h with shaking (100 rpm) at 37 ± 0.5 °C. Then material was moved to empty wells, rinsed three times, and stained for 20 min with the fluorescent dyes: SYTO 9 (λ_exc_ = 543 nm) and propidium iodide (λ_exc_ = 543 nm) (both at the concentration of 1 µL/mL) (LIVE/DEAD™ *Bac*Light™ Bacterial Viability Kit, for microscopy, Invitrogen™, Thermo Fisher Scientific; Waltham, MA, USA), rinsed and observed in confocal microscope with 2× digital magnification (Olympus IX83 Fluoview FV 1200 (Tokyo, Japan), magnification 20×).

Four types of hydrogels (pure, with hydroxyapatite, with fluconazole as well as with hydroxyapatite and fluconazole) were used to compare biofilm formation ability. Material was aseptically cut into pieces and put into 24-well plate. The masses of particular samples were as follows: 10 mg of pure 6-Anhydro-α-l-Galacto-β-d-Galactan, 20 mg of 6-Anhydro-α-l-Galacto-β-d-Galactan with apatite, 20 mg of 6-Anhydro-α-l-Galacto-β-d-Galactan with fluconazole and 30 mg of 6-Anhydro-α-l-Galacto-β-d-Galactan with hydroxyapatite and fluconazole in order to maintain equal concentration of the particular components in each. Fungal suspensions were prepared as described above and 1 mL of each strain suspension was added to the wells. The plate was incubated for 24 h with shaking (100 rpm) at 37 ± 0.5 °C. Then material was moved to empty wells, rinsed three times, and stained for 20 min with the fluorescent dyes: SYTO 9 (λ_exc_ = 543 nm) and propidium iodide (λ_exc_ = 543 nm) (both at the concentration of 1 µL/mL) (LIVE/DEAD™ *Bac*Light™ Bacterial Viability Kit, for microscopy, Invitrogen™, Thermo Fisher Scientific; Waltham, MA, USA), rinsed and observed in confocal microscope with 2× digital magnification (Olympus IX83 Fluoview FV 1200 (Tokyo, Japan), magnification 20×).

#### 4.7.4. Influence of Hydrogels on Planktonic Cells

After the incubation of fungal suspensions in the presence of hydrogels as was described previously (see confocal microscopy), the absorbance of the fungal cultures above the materials was measured. One hundred µL fluid samples were gently transferred to empty wells and the absorbance (OD 600 nm) was measured using microplate reader (Varioskan LUX, Thermo Fisher Scientific, Waltham, MA, USA). The fungal cultures incubated without hydrogels were used as the control samples.

## 5. Conclusions

The XRPD and FT-IR analysis have shown that all components of desired hydrogels remained unchanged after the process of their preparation. On the basis of drug release tests, it can be concluded that the entire fluconazole is released in a short time (approx. 1 h). Fluconazole released from hydrogels effectively eradicated planktonic yeast cells; however, no significant effect against biofilms was observed, with the exception of *R. mucilaginosa*. Further studies towards potential application of prepared formulations are required to obtain desirable antifungal effects.

## Figures and Tables

**Figure 1 ijms-22-03112-f001:**
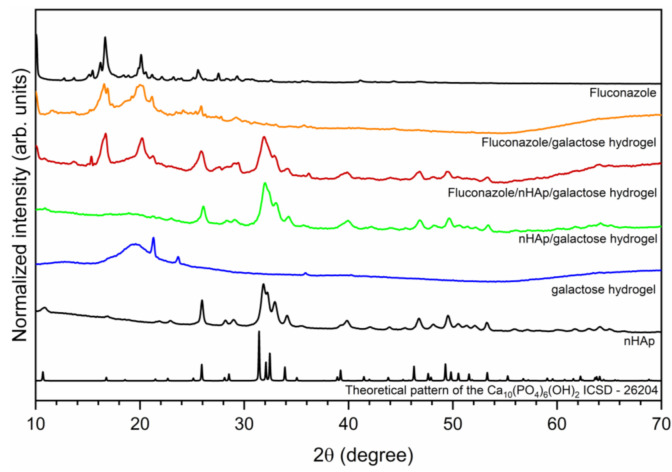
X-ray powder diffraction pattern of sintered nanohydroxyapatite, pure 6-Anhydro-α-l-Galacto-β-d-Galactan hydrogel (galactose hydrogel), nHAp/6-Anhydro-α-l-Galacto-β-d-Galactan hydrogel, Fluconazole/nHAp/6-Anhydro-α-l-Galacto-β-d-Galactan hydrogel, Fluconazole/6-Anhydro-α-l-Galacto-β-d-Galactan hydrogel and fluconazole.

**Figure 2 ijms-22-03112-f002:**
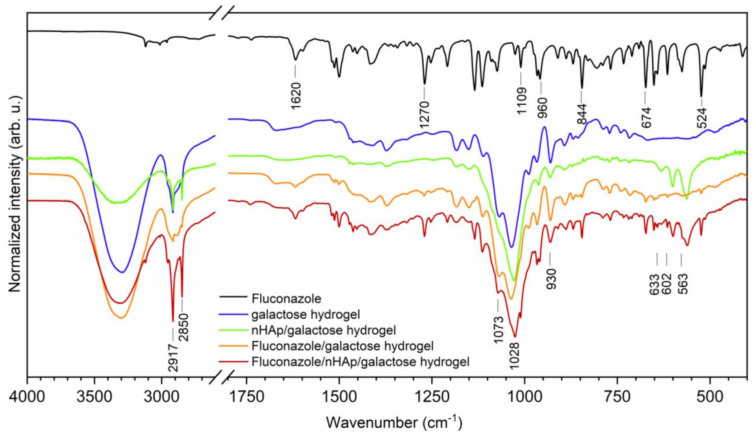
The Fourier-Transform Infrared spectra of Fluconazole, pure 6-Anhydro-α-l-Galacto-β-d-Galactan hydrogel (galactose hydrogel), nHAp/6-Anhydro-α-l-Galacto-β-d-Galactan hydrogel, Fluconazole/nHAp/6-Anhydro-α-l-Galacto-β-d-Galactan hydrogel, Fluconazole/6-Anhydro-α-l-Galacto-β-d-Galactan hydrogel.

**Figure 3 ijms-22-03112-f003:**
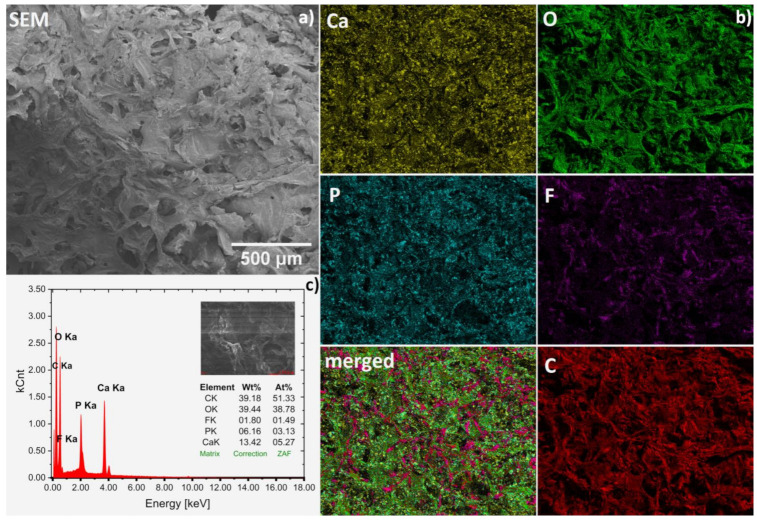
Representative SEM image (magnification 100×) (**a**) with SEM-EDS elemental mapping (**b**) of the Fluconazole/nHAp/6-Anhydro-α-l-Galacto-β-d-Galactan hydrogel together with EDS spectrum (**c**) for average area.

**Figure 4 ijms-22-03112-f004:**
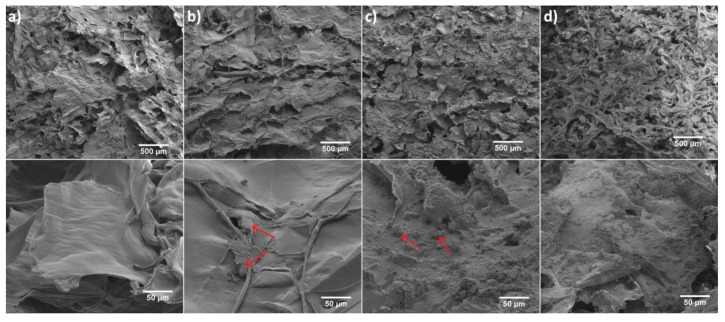
SEM images of (**a**) 6-Anhydro-α-l-Galacto-β-d-Galactan hydrogel, (**b**) Fluconazole/6-Anhydro-α-l-Galacto-β-d-Galactan hydrogel (red arrows indicate example areas with fluconazole). (**c**) nHAp/6-Anhydro-α-l-Galacto-β-d-Galactan hydrogel (red arrows indicates example areas with hydroxyapatite) and (**d**) Fluconazole/nHAp/6-Anhydro-α-l-Galacto-β-d-Galactan hydrogel. Magnification 100× (scale bar = 500 μm) and 1000× (scale bar = 50 μm).

**Figure 5 ijms-22-03112-f005:**
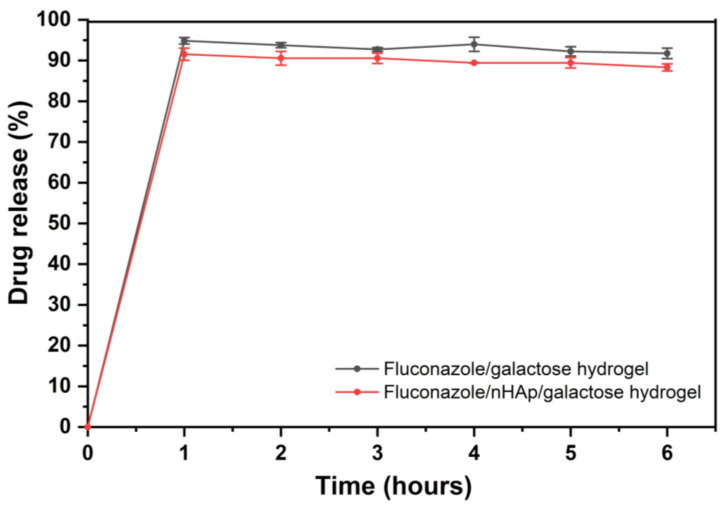
Time-dependent fluconazole release from the Fluconazole/6-Anhydro-α-l-Galacto-β-d-Galactan hydrogel (galactose hydrogel) and from Fluconazole/nHAp/6-Anhydro-α-l-Galacto-β-d-Galactan hydrogel presented as a % released.

**Figure 6 ijms-22-03112-f006:**
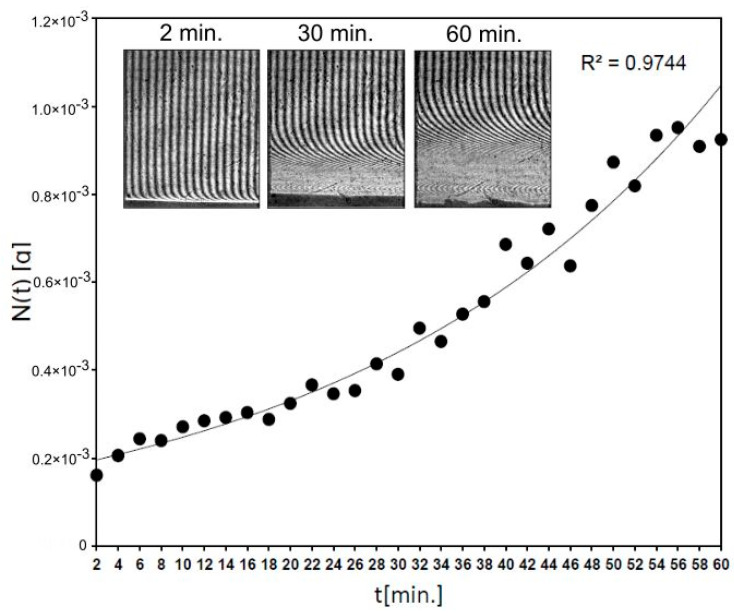
The kinetics of fluconazole release from 6-Anhydro-α-l-Galacto-β-d-Galactan hydrogel modified with nHAp measured by laser interferometry system. The example of interferograms (after times 2, 30 and 60 min.) are presented.

**Figure 7 ijms-22-03112-f007:**
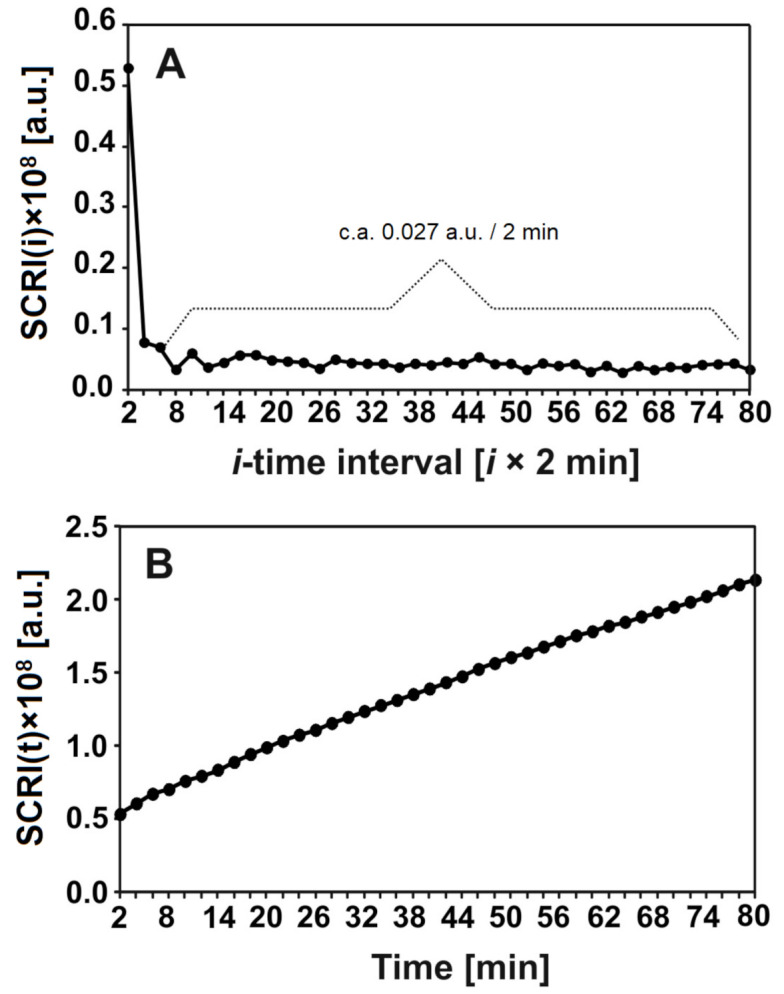
The amount (a.u.) of released hydrogel degradation products after incubation with lysozyme at 1 mg/mL for 18 h at 37 °C in time intervals = 2 min (**A**) or after time t (**B**).

**Figure 8 ijms-22-03112-f008:**
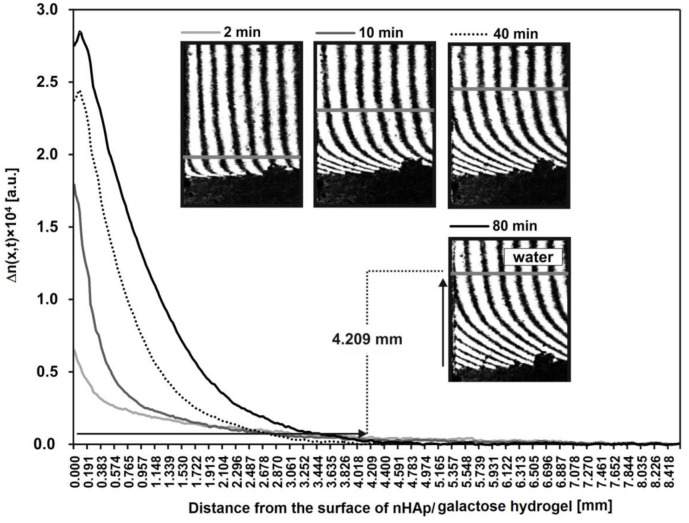
The spatiotemporal distributions of refraction index changes for the hydrogel degradation products after incubation with lysozyme at 1 mg/mL (2, 10, 40 and 80 min). These distributions illustrate the spatiotemporal concentration distributions in arbitrary units. As an example, the distance of the transported hydrogel degradation products after 80 min is presented on interferogram and curve.

**Figure 9 ijms-22-03112-f009:**
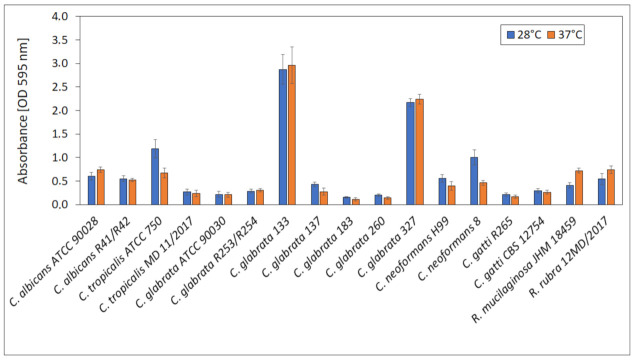
Biofilm formation at 28 and 37 °C.

**Figure 10 ijms-22-03112-f010:**
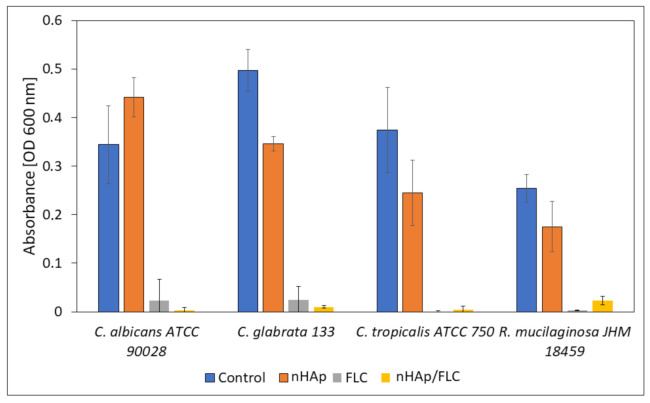
The absorbance measurement (OD 600 nm) of the fungal cultures incubated in the presence of hydrogels (nHAp- nanohydroxyapatite; FLC-fluconazole) compared with the control sample.

**Figure 11 ijms-22-03112-f011:**
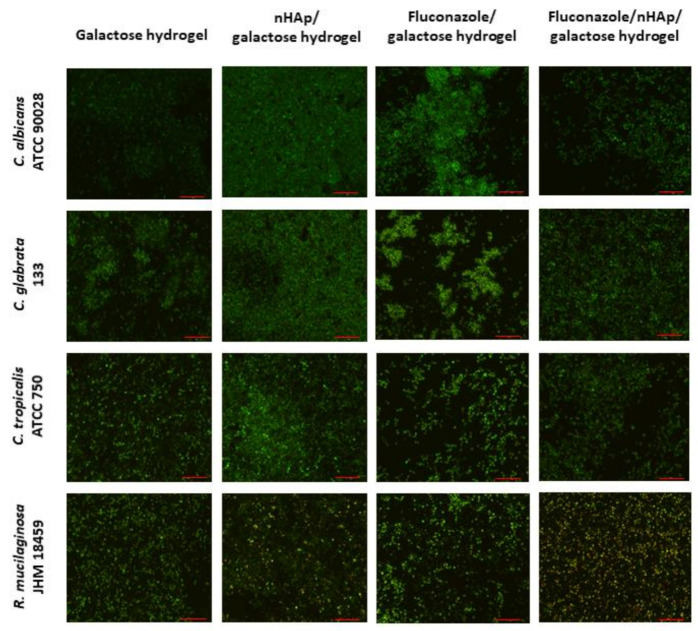
Fungal biofilm on the surface of 6-Anhydro-α-l-Galacto-β-d-Galactan (galactose hydrogel), 6-Anhydro-α-l-Galacto-β-d-Galactan with nanohydroxyapatite (nHAp/galactose hydrogel), 6-Anhydro-α-l-Galacto-β-d-Galactan with fluconazole (Fluconazole/galactose hydrogel) and 6-Anhydro-α-l-Galacto-β-d-Galactan with nanohydroxyapatite and fluconazole (Fluconazole/nHAp/galactose hydrogel) (scale bar = 50 µm).

**Table 1 ijms-22-03112-t001:** Minimal inhibitory concentrations [µg/mL] of fluconazole against tested fungal strains.

Strain		MIC (µg/mL)
*C. albicans*	ATCC 90028	<2
	R41/R42	<2
*C. tropicalis*	ATCC 750	4
	11 MD/2017	15.6
*C. glabrata*	ATCC 90030	250
	R253/R254	31.25
	133	<2
	137	62.5
	183	<2
	260	125
	327	<2
*C. neoformans*	H99	62.5
	No. 8	15.6
*C. gatti*	R265	62.5
	CBS 12754	125
*R. mucilaginosa*	JHM 18459	>250
*R. rubra*	12MD/2017	>250

## Data Availability

Not applicable.

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
