# Peer review of "New Approach to Antifungal Activity of Fluconazole Incorporated into the Porous 6-Anhydro-α-l-Galacto-β-d-Galactan Structures Modified with Nanohydroxyapatite for Chronic-Wound Treatments—In Vitro Evaluation"

_ijms, 2021, doi:10.3390/ijms22063112_

Round 1
Reviewer 1 Report
Dear authors,
The subject is interesting and original; the research very well conducted.
I suggest discussing the impact of fungal biofilms on wound healing. Most of the reports suggest the implication of bacteria and there are few on fungi. You wrote that biofilms can alter normal wound healing, but it is still not clear to me how fungal biofilms have this effect- more references needed in the article to argue the importance of antifungal therapy.
The article needs significant editing and an English Grammar professional check.
There are some errors in the text regarding two references- please correct.
Author Response
Dear Editor,
We would like to express our sincerest gratitude to the Reviewers for their enormous efforts in criticizing the manuscript. We have taken into account all raised question here follows the detailed answers to the Reviewers. Moreover, all changes we have made to the original manuscript, are marked in the red colour in the text.
Reviewer 1:
The subject is interesting and original; the research very well conducted.
I suggest discussing the impact of fungal biofilms on wound healing. Most of the reports suggest the implication of bacteria and there are few on fungi. You wrote that biofilms can alter normal wound healing, but it is still not clear to me how fungal biofilms have this effect- more references needed in the article to argue the importance of antifungal therapy.
Answer: Thank you for the comment. We agree that these data are not sufficiently outlined, thus we extended the discussion with more information. The studies are focused mainly on the effects of bacterial biofilms, which in fact delay wound healing by the cycles of re-infections when cells disperse from biofilms to colonize new sites, or by enhancing wound exudates. The studies on fungi infecting diabetic foot ulcers also confirm interference with the healing process, mainly by causing tissue necrosis. We added the information and the references in the Discussion.
The article needs significant editing and an English Grammar professional check.
Answer: Thank you for pointing that out. We thoroughly checked the grammar and corrected the mistakes. Also style errors were corrected (like units and spelling mistakes)
There are some errors in the text regarding two references- please correct.
Answer: Thank you. We corrected the references.

Reviewer 2 Report
The study performed by Rewak-Soroczynska et al. provides an interesting and novel subject in the antimicrobial therapy field. The combination of organic polymer-based hydrogels with calcium phosphate nanoparticles for wound treatment is of significant interest at the moment.
I believe that there are some minor revisions that must be considered before publishing.
Lines 89 and 106 - Error! Reference source not found. Please solve the issue.
Line 163 - What does BCS stand for?
Chapter 2.2 - Dissolution profiles and drug release from hydrogels are two different tests. Please correct the issue.
Antifungal effects of nHap should be compared with other types of nanoparticles, such as silver, gold, or iron oxide. Thus, I suggest discussing the following references:
- 10.33263/briac106.65876596
- 10.1186/s13568-019-0857-7
- 10.1155/2020/9535432
- 10.33263/briac101.902907
Additionally, English must be thoroughly revised, as there are many grammatical errors throughout the manuscript.
Author Response
Dear Editor,
We would like to express our sincerest gratitude to the Reviewers for their enormous efforts in criticizing the manuscript. We have taken into account all raised question here follows the detailed answers to the Reviewers. Moreover, all changes we have made to the original manuscript, are marked in the red colour in the text.
Reviewer 2:
The study performed by Rewak-Soroczynska et al. provides an interesting and novel subject in the antimicrobial therapy field. The combination of organic polymer-based hydrogels with calcium phosphate nanoparticles for wound treatment is of significant interest at the moment.
I believe that there are some minor revisions that must be considered before publishing.
Lines 89 and 106 - Error! Reference source not found. Please solve the issue.
Answer: Thank you for the comment. We corrected the errors.
Line 163 - What does BCS stand for?
Answer: We added this information in the manuscript.
Chapter 2.2 - Dissolution profiles and drug release from hydrogels are two different tests. Please correct the issue.
Answer: Thank you for the comment. The both tests were related to drug release. It has been corrected.
Antifungal effects of nHap should be compared with other types of nanoparticles, such as silver, gold, or iron oxide. Thus, I suggest discussing the following references:
- 33263/briac106.65876596
- 1186/s13568-019-0857-7
- 1155/2020/9535432
- 33263/briac101.902907
Answer: Thank you for the suggestion. We discussed suggested references in the Discussion. We believe that it elevated manuscript value and hope it would be sufficient.
Additionally, English must be thoroughly revised, as there are many grammatical errors throughout the manuscript.
Answer: We revised the language and corrected grammar and style mistakes.
